# Leptospiral LPS escapes mouse TLR4 internalization and TRIF-associated antimicrobial responses through O antigen and associated lipoproteins

Delphine Bonhomme[1,2], Ignacio Santecchia[1,2], Frédérique Vernel-Pauillac[1], Martine Caroff[3], Pierre Germon[4], Gerald Murray[5], Ben Adler[5], Ivo G. Boneca[1], Catherine Werts[1] *

1 Institut Pasteur, Unité Biologie et Génétique de la Paroi Bactérienne, Paris, France; CNRS, UMR 2001 « Microbiologie intégrative et Moléculaire », Paris, France; INSERM, Equipe Avenir, Paris, France, 2 Université de Paris, Sorbonne Paris Cité, Paris, France, 3 LPS-BioSciences, Université de Paris-Saclay, Orsay, France, 4 INRAE, UMR ISP, Université François Rabelais de Tours, Nouzilly, France, 5 Department of Microbiology, Biomedicine Discovery Institute, Monash University, Melbourne, Australia

* cwerts@pasteur.fr

**Data Availability Statement:** All relevant data are within the manuscript and its Supporting Information files.

## Abstract

Leptospirosis is a worldwide re-emerging zoonosis caused by pathogenic *Leptospira* spp. All vertebrate species can be infected; humans are sensitive hosts whereas other species, such as rodents, may become long-term renal carrier reservoirs. Upon infection, innate immune responses are initiated by recognition of Microbial Associated Molecular Patterns (MAMPs) by Pattern Recognition Receptors (PRRs). Among MAMPs, the lipopolysaccharide (LPS) is recognized by the Toll-Like-Receptor 4 (TLR4) and activates both the MyD88-dependent pathway at the plasma membrane and the TRIF-dependent pathway after TLR4 internalization. We previously showed that leptospiral LPS is not recognized by the human-TLR4, whereas it signals through mouse-TLR4 (mTLR4), which mediates mouse resistance to acute leptospirosis. However, although resistant, mice are known to be chronically infected by leptospires. Interestingly, the leptospiral LPS has low endotoxicity in mouse cells and is an agonist of TLR2, the sensor for bacterial lipoproteins. Here, we investigated the signaling properties of the leptospiral LPS in mouse macrophages. Using confocal microscopy and flow cytometry, we showed that the LPS of *L. interrogans* did not induce internalization of mTLR4, unlike the LPS of *Escherichia coli*. Consequently, the LPS failed to induce the production of the TRIF-dependent nitric oxide and RANTES, both important antimicrobial responses. Using shorter LPS and LPS devoid of TLR2 activity, we further found this mTLR4-TRIF escape to be dependent on both the co-purifying lipoproteins and the full-length O antigen. Furthermore, our data suggest that the O antigen could alter the binding of the leptospiral LPS to the co-receptor CD14 that is essential for TLR4-TRIF activation. Overall, we describe here a novel leptospiral immune escape mechanism from mouse macrophages and hypothesize that the LPS altered signaling could contribute to the stealthiness and chronicity of the leptospires in mice.

**Funding:** This study received funding from the French Government's Investissement d'Avenir program, Laboratoire d'Excellence "Integrative Biology of Emerging Infectious Diseases" (grant nˆANR 10 LABX 62 IBEID to IB). DB received funding from the Ecole Doctorale Frontières de l'Innovation en Recherche et Education (FIRE), Programme Bettencourt. IS was supported by the Institut Carnot Pasteur Microbes & Santé given to the Pasteur-Paris University PhD program and the "Fin de thèse de science" number FDT201805005258 granted by "Fondation pour la recherche médicale (FRM)". The funders had no role in study design, data collection and analysis, decision to publish, or preparation of the manuscript.

**Competing interests:** The authors have declared that no competing interests exist.

## Author summary

*Leptospira interrogans* is a bacterial pathogen, responsible for leptospirosis, a worldwide neglected reemerging disease. *L. interrogans* may cause an acute severe disease in humans, whereas rodents and other animals asymptomatically carry the leptospires in their kidneys. They can therefore excrete live bacteria in urine and contaminate the environment. Leptospires are stealth pathogens known to escape the innate immune defenses of their hosts. They are covered in lipopolysaccharide (LPS), a bacterial motif recognized in mammals through the Toll-like receptor 4 (TLR4), which triggers two different signaling pathways. We showed previously that pathogenic leptospires fully escape TLR4 recognition in humans. Here we focused on the LPS signaling in mice that are, although resistant to acute leptospirosis, chronically infected. We showed in mouse cells that the leptospiral LPS triggers only one arm of the TLR4 pathway and escapes the other, hence avoiding production of antimicrobial compounds. Removing the lipoproteins that always co-purify with the leptospiral LPS, or using shorter LPS, restores the stimulation of both pathways. This suggests a novel escape mechanism linked to the LPS and involving lipoproteins that could be instrumental for leptospires to escape the mouse defense and to allow for their chronic renal colonization.

## Introduction

*Leptospira* spp. are the causative agents of leptospirosis, a neglected global zoonosis. Leptospirosis is currently re-emerging due to global warming and causes about 60,000 deaths per year worldwide [1]. Infection by pathogenic species such as *L. interrogans* in humans may often cause asymptomatic or "flu-like" infections, but can also cause severe disease with a fatality rate of up to 10%. Notably, it may also be an overlooked cause of chronic kidney disease [2]. All other vertebrates can be infected, with symptoms varying from one species to another. For instance, infection may cause morbidity and abortion in cattle or uveitis in horses [3]. Rodents such as mice and rats play an important role in the endozoonotic cycle of the disease because they are resistant to acute infection and can, like wild animals or cattle, become lifelong reservoirs of these bacteria [3]. Upon infection with *Leptospira*, the proximal renal tubules may become chronically infected leading to excretion of bacteria in urine and transmission of the disease [4,5]. Unlike most other spirochetes, like *Borrelia burgdorferi* and *Treponema pallidum*, respective agents of Lyme disease and syphilis, *Leptospira* spp. possess a lipopolysaccharide (LPS) as one of their outer membrane components [6]. This LPS is an essential molecule for the bacterial outer membrane integrity and is known to contribute to leptospiral virulence [7]. The LPS is composed of a hexa-acylated lipid A, allowing anchoring in the outer membrane, and a succession of sugar units distributed between inner core, outer core and O antigen. It is the molecular basis for classification of *Leptospira* into more than 300 different serovars [8]. Furthermore, leptospiral LPS has been described as atypical in terms of structure. First, the lipid A possesses the following unconventional structures: (i) non-saturated fatty acid chains; (ii) a 2,3-diamino-2,3-dideoxy-D-glucose (DAG) disaccharide backbone carrying 4 primary amide-linked fatty acids; (iii) a methylated 1-phosphate group and (iv) no 4'-phosphate group [9]. The lack of reactive phosphate groups is most probably responsible for the lack of human TLR4 recognition [10]. Additionally, silver stained electrophoretic profiles of LPS from pathogenic leptospiral strains were shown to be different from that of *E. coli* LPS [11] and the carbohydrates composing the O antigen were described to be more irregular and complex than the

classical repeated units pattern [12]. The leptospiral O antigen LPS biosynthesis *rfb* locus was found to be between 4 to 120 kb [13–15] long, depending on the species and strains. In *L. interrogans* and *L. borgpetersenii* it was described to include at least 30 open reading frames [16,17]. Among the various genes of the locus, glycosyl transferases and integral membrane proteins involved in full-length LPS transport were identified as well as enzymes for rhamnose sugar biosynthesis [17]. However, most of these open reading frames encode proteins of unknown function and the overall structure of the O antigen section of the leptospiral LPS remains unknown. Interestingly, LPS of saprophytic species of *Leptospira* was found to be short, although not completely rough [11].

Molecular Associated Microbial Patterns (MAMPs) are conserved molecules among microbes that are recognized by the host innate immune system *via* Pattern Recognition Receptors (PRRs). PRRs can be soluble or expressed in cells, either membrane-bound or cytosolic. Activation of these cellular PRRs leads to the production of pro-inflammatory cytokines and chemokines that are the signals that orchestrate the immune response [18]. Among these PRRs, the plasma membrane Toll-Like-Receptor 4 (TLR4) and Toll-Like-Receptor 2 (TLR2) sense bacterial LPS and bacterial lipoproteins, respectively. Subsequent activation of signaling pathways *via* the adaptors MyD88/TIRAP leads to the early nuclear translocation of the transcription factor NF-κB, a central regulator of inflammatory and antimicrobial responses. In addition, upon LPS stimulation, TLR4 is also endocytosed. Signaling from endosomal TLR4 relies on the TRIF/TRAM adaptors and induces late NF-κB translocation as well as IRF3 activation, leading to the production of IFN-β, and interferon stimulating genes [19]. Many cofactors play essential roles in the activation of TLR4 by bacterial LPS, including the LPS-binding protein (LBP) and the co-receptor cluster of differentiation 14 (CD14) that is either soluble or attached by a glycosylphosphatidylinositol (GPI) anchor on the surface of mononuclear phagocytes [20]. Many studies have highlighted the important roles of CD14 for the efficient signaling of the LPS, and it was shown that it is important for TLR4 internalization and TRIF signaling [19,21,22].

The LPS of *L. interrogans* is unconventional in its recognition by the PRRs of the innate immune system. First, it is differentially recognized by the human and mouse TLR4. Leptospiral LPS activates the mouse, but not the human TLR4 [23]. In addition, in mouse cells leptospiral LPS has very low endotoxic activity compared to other bacterial LPS such as from *Escherichia coli* [23,24]. Another signaling particularity of the leptospiral LPS is that it activates TLR2, both in human and mouse cells [24,25]. This immunological property is independent of the Lipid A moiety [23] and could be attributed to co-purifying lipoproteins. It has been described that the co-receptor CD14 is needed for this TLR2 activation [24]. Of note, these co-purifying lipoproteins resist the classical hot water/phenol extraction and mild proteinase K treatment [24], suggesting strong interaction with the LPS. There are more than 170 open reading frames encoding putative lipoproteins in the genome of *L. interrogans* [26]; most of them are of unknown function. Among these, LipL32 is the most abundant lipoprotein. It was shown not to be essential for virulence [27] and to be a TLR2-TLR1 agonist [23,24,28]. On the other hand, Loa22, another abundant lipoprotein was shown to be essential for leptospiral virulence [29]. Even though most of the functions of these lipoproteins remain unknown, previous publications showed that some leptospiral lipoproteins play a role in the innate immune escape. Indeed, LipL21 binds tightly to the leptospiral peptidoglycan and hence prevents its recognition by the PRRs NOD1 and NOD2 [30]. Additionally, LipL21 and other lipoproteins have also been described as inhibitors of neutrophil myeloperoxidase [31].

To our knowledge, the mechanism underlying the low endotoxicity of the leptospiral LPS seen in mouse cells [23,24,32] has never been investigated. Moreover, the role of the co-purifying lipoproteins that often contaminate leptospiral LPS preparations and confer their TLR2 activity

remains to be studied. Hence, in this study we aimed to characterize the atypical signaling of the leptospiral LPS in murine cells and to elucidate the role of these co-purifying lipoproteins.

## Results

### Leptospiral LPS does not trigger mouse-TLR4 internalization

Upon stimulation with bacterial LPS, it was described that surface TLR4 is internalized [22] within the first hour. Therefore, we first assessed whether that was the case upon stimulation with leptospiral LPS. Surprisingly, immunofluorescence analysis of RAW264.7 cells stimulated for 1 h with LPS of the pathogenic *L. interrogans* strain Verdun revealed that mouse-TLR4 (mTLR4) remained localized on the plasma membrane, in contrast to stimulation with *E. coli* LPS, after which mTLR4 localization was intracellular (Fig 1A). The mTLR4 specificity of the anti-TLR4 antibody was confirmed using TLR4 KO bone marrow derived macrophages (BMDMs) (S1A Fig). mTLR4 fluorescence profiles were further quantified on cross-sections (S1B Fig) performed on at least 40 cells and were averaged to estimate the localization of the mTLR4 receptor upon stimulation. These quantifications confirmed that the plasma membrane localization of mTLR4, characterized by a two-peak distribution of the fluorescence profile, was modified only upon stimulation with *E. coli* LPS, but not *L. interrogans* LPS (Fig 1B). This phenotype was confirmed with flow cytometry analyses and surface mTLR4 staining on stimulated RAW264.7 cells. We observed that the mean fluorescence intensity (MFI), corresponding to surface mTLR4, decreased only upon stimulation with LPS of *E. coli* (Fig 1C, left panel and S1D Fig). Given that leptospiral serovars have different LPS, we further showed that mTLR4 internalization was not induced by LPS of any of the 3 main serovars of *L. interrogans* (Icterohaemorrhagiae strain Verdun; Copenhageni strain Fiocruz L1-130; Manilae strain L495) (Fig 1C, right panel). This was consistent with the similar LPS profiles observed between the different serovars after silver staining (S1C Fig). To exclude the possibility that the LPS of *L. interrogans* could induce a delayed internalization of mTLR4, we performed cytometry analyses at different time points and showed that there was no delayed internalization of the leptospiral LPS, even after 24 h of stimulation (S1E Fig). Furthermore, to assess a potential effect of the dose of LPS on mTLR4 internalization, we showed that the leptospiral LPS did not trigger mTLR4 internalization at either 100 ng/mL or 1 µg/mL (S1F Fig).

### Leptospiral LPS avoids TRIF responses but activates MyD88 at high concentrations

Since leptospiral LPS was unable to trigger internalization of mTLR4, we then investigated whether either MyD88- (plasma membrane adaptor) or TRIF- (endosome adaptor) dependent responses were induced 24 h post stimulation. We first showed, using WT and MyD88 KO BMDMs that the production of the KC chemokine was fully dependent on the MyD88 adaptor, with no difference in the production induced by LPS of *L. interrogans* or LPS of *E. coli* (Fig 2A, left panel). Consistent with previous studies [24], we confirmed that the KC production upon leptospiral LPS stimulation was dependent on both TLR4 and TLR2 (S2A Fig), and we confirmed using TLR2/TLR4 DKO BMDMs that only TLR2 and TLR4 contribute to LPS signaling in the case of both *L. interrogans* and *E. coli* LPS (S2B Fig), consistent with our previous work [23]. On the other hand, we showed that the production of RANTES was largely dependent on the TRIF adaptor pathway (S2C Fig) and that the LPS of *L. interrogans* induced around 3 times less RANTES than the LPS of *E. coli* (Fig 2A). Furthermore, the production of nitric oxide (NO), a potent antimicrobial compound [33], which we also showed to be fully dependent on the TRIF adaptor (S2C Fig), was almost completely blunted upon stimulation

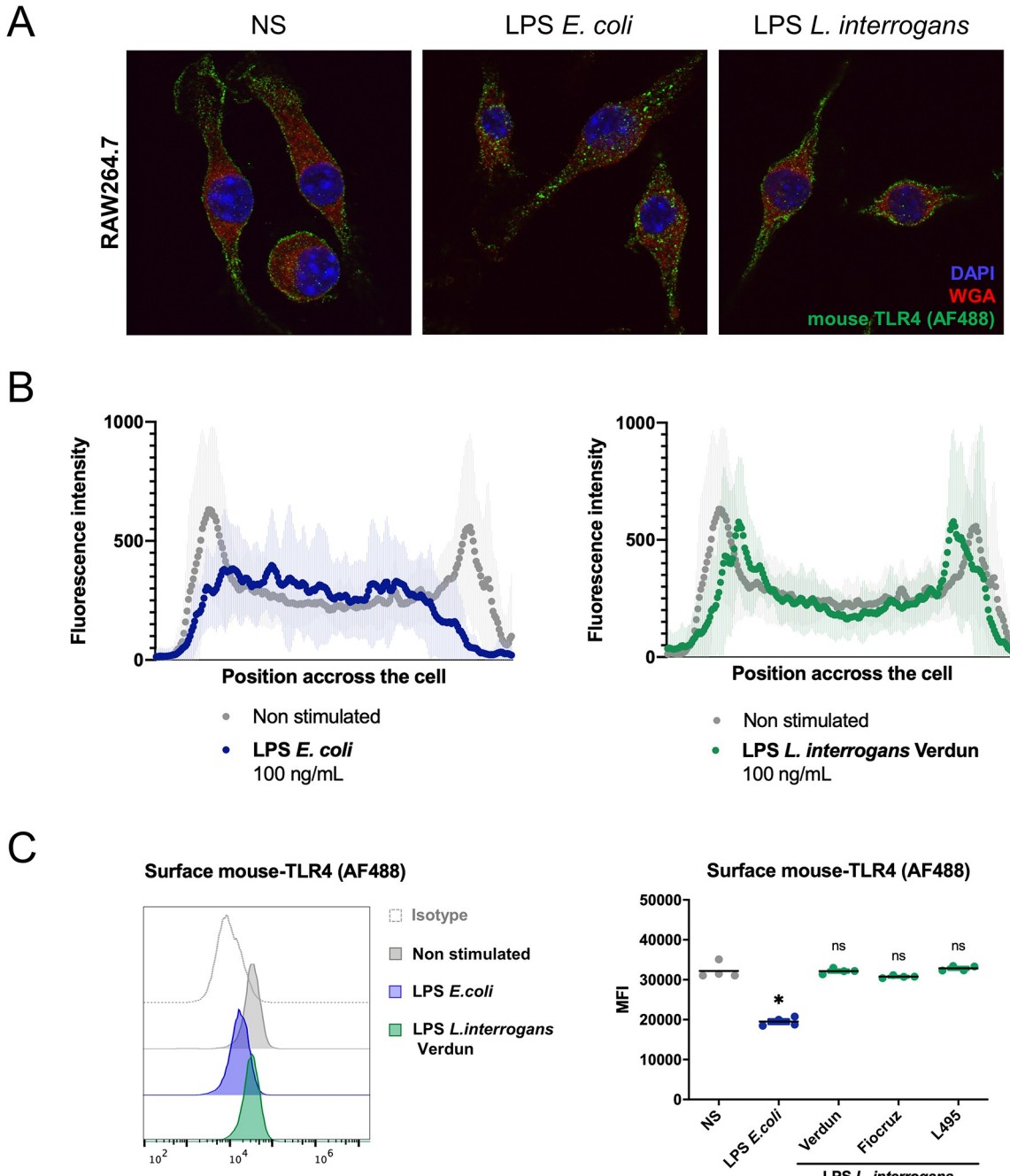

**Fig 1. Leptospiral LPS does not trigger TLR4 internalization. A)** Representative images of confocal IF analyses of RAW264.7 macrophage-like cells upon 1h stimulation with 100 ng/mL of LPS of *Escherichia coli* or LPS of *Leptospira interrogans* and staining for mouse-TLR4 (mTLR4), glycoproteins (Weat Germ Agglutinin, WGA) and nuclei (DAPI). **B)** Average mTLR4 fluorescence profiles quantified on cross-section of the cells (see S1B Fig for more details) either non-stimulated (grey), stimulated with LPS of *E. coli* (blue) or LPS of *L. interrogans* (green). Data are represented as mean of > 40 cells and shades correspond to +/- SD. **C)** Flow cytometry analysis of surface mTLR4 (or isotype) on RAW264.7 cells upon 1h stimulation with 1μg/mL of LPS of *E. coli* (blue) or *L. interrogans* (green) (left panel) and corresponding mean fluorescence intensities (MFI) (right panel) with various serovars of *L. interrogans* LPS (serovar Icterohaemorrhagiae strain Verdun, serovar Copenhageni strain Fiocruz L1-130, serovar Manilae strain L495) (see S1D Fig for complete gatings). Each dot of MFI corresponds to the analysis of one well (10 000–30 000 events). All experiments shown are representative of at least 3 independent experiments and statistical analyses were performed using the non-parametric Mann-Whitney test.

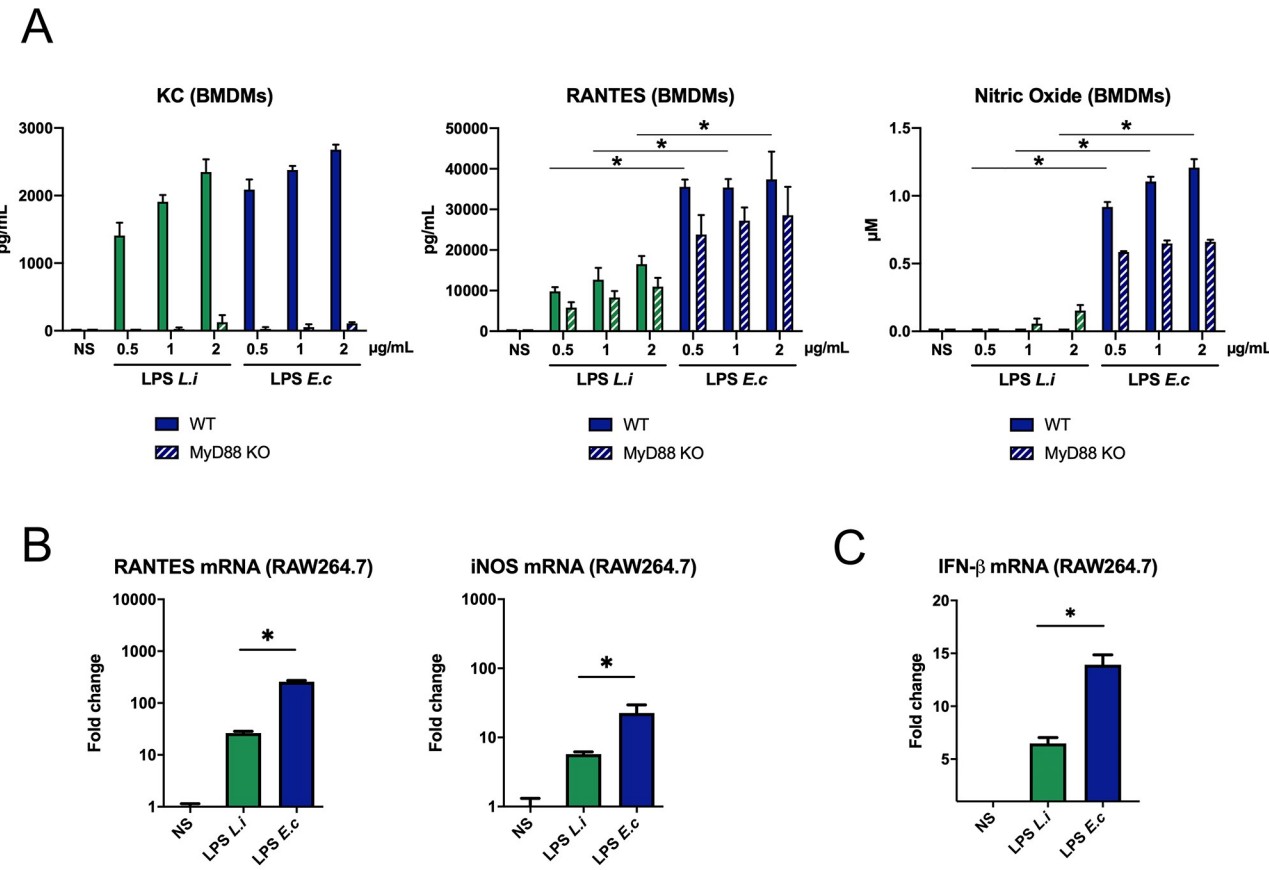

**Fig 2. Leptospiral LPS avoids TRIF-dependent responses but activates MyD88. A)** Production of KC, RANTES and NO by WT and MyD88 KO BMDMs after 24h stimulation with 500 ng/mL, 1 µg/mL and 2 µg/mL of LPS of *L. interrogans* (*L.i*, green) or LPS of *E. coli* (*E.c*, blue). **B)** mRNA levels of RANTES, iNOS and **C)** IFN-β in RAW264.7 cells after stimulation with 1 µ/mL of LPS of *L. interrogans* (*L.i*, green) or LPS of *E. coli* (*E.c*, blue). Data are represented as mean (+/- SD) of *n* = 3/4 technical replicates and are representative of at least 3 independent experiments. Statistical analyses were performed using the non-parametric Mann-Whitney test.

with the leptospiral LPS, while it was elicited in a dose dependent manner by the LPS of *E. coli* (Fig 2A). In addition, both RANTES and iNOS were analyzed at the transcriptional level in RAW264.7 cells. RT-qPCR revealed that the leptospiral LPS induced much lower TRIF-dependent responses than *E. coli* LPS (Fig 2B). Similarly, we analyzed the levels of IFN-β mRNA and our results also showed that the LPS of *L. interrogans* induced lower production of IFN-β mRNA, compared to the LPS of *E. coli* (Fig 2C). We further controlled that the IFN-β mRNA levels were representative of the cytokine production and that the induction of IFN-β was dependent on the adaptor TRIF (S2D Fig). These results indicated that leptospiral LPS avoids TRIF-dependent, but not MyD88-dependent, responses. Surprisingly, we showed that at lower concentration, leptospiral LPS avoided not only the TRIF-dependent NO production, but also the production of MyD88-dependent KC (S2E Fig). These results suggest that at high concentrations, the leptospiral LPS escapes the mTLR4 internalization and specific TRIF-dependent responses, whereas at low concentrations, the leptospiral LPS escapes both pathways.

## Shorter leptospiral LPS induces more TLR4-TRIF responses

Because we showed that the leptospiral LPS could induce signaling *via* the TLR4-MyD88 pathway, we hypothesized that the TRIF-escape phenotype we observed was not due to a poor

interaction of the lipid A with mTLR4. Instead, we hypothesized that the O antigen of the LPS could be involved in the TRIF escape. Therefore, we investigated whether different leptospiral LPS could induce the internalization of mTLR4 and trigger responses *via* the TRIF adaptor. We first studied the LPS from *L.interrogans* Manilae strain M895, an attenuated strain that produces a shorter LPS. This LPS was compared to the LPS of the WT strain Manilae L495 after quantification of lipid A acyl chains and estimation of their respective molar concentration. We confirmed the truncation of this LPS, which showed a downward shift of the upper band visible on silver staining (Fig 3A). Immunofluorescence analyses of RAW264.7 cells stimulated for 1 h by this shorter M895 LPS revealed that the localization of mTLR4 was slightly modified upon stimulation, as quantified by the reduction in the two peaks corresponding to the membrane associated mTLR4 and the increase on the internalized signal (Fig 3B), although some mTLR4 remained visible on the membrane of the cells (Fig 3C). Furthermore, we showed that this shorter LPS induced more RANTES and more NO production (TRIF-dependent responses) than the same molar concentration of the full-length leptospiral LPS (Fig 3D). Analyses of the mRNA levels of RANTES, iNOS and IFN-β confirmed that the short LPS of the M895 strain triggers more TRIF-dependent responses that the full-length L495 leptospiral LPS (Fig 3E and 3F).

In addition, we also purified and characterized the LPS of the saprophytic *L. biflexa* (*L.b*) serovar Patoc strain Patoc and showed that it was much shorter than the LPS of the pathogenic species (S3A Fig). Because the structure of the Patoc Lipid A is unknown, we could not estimate its exact molar concentration. Nevertheless, based on same LPS amount by weight, HEK reporter system analyses showed that both Patoc and Verdun LPS had similar mouse-TLR4 activity in comparison with internal control (LPS of *E. coli*) (S3B Fig), whereas the LPS of Patoc had no TLR2 activity, in contrast to Verdun LPS (S3B Fig). Noteworthy, no differential activities between the human and mouse-TLR2 receptors were observed, hence allowing us to work indifferently with either one of these receptors to monitor the TLR2 activity of our LPS preparations. We further confirmed by analyses of TLR4 KO and TLR2 KO BMDMs that the signaling of Patoc LPS was TLR2-independent (S3C Fig). Consistent with the results of the M895 shorter LPS, we showed that the short Patoc LPS induced in macrophages a higher production of RANTES and NO than the Verdun LPS (S3D Fig). Altogether these data strongly suggest that the full-length intact LPS is important to avoid mTLR4 internalization and subsequent TRIF-dependent responses.

## Soluble CD14 participates in the enhanced signaling of the shorter LPS

After investigating the bacterial components that could be involved, we decided to assess whether host factors could also be implicated in the mechanism. We first investigated the role of serum in the signaling. Both full-length and shorter leptospiral LPS benefited from serum addition for RANTES production (S4 Fig). Next, we investigated whether serum components could contribute to the differential signaling observed between the full-length and the shorter LPS. CD14 appeared as a good candidate given its role in TLR4 endocytosis [22] and in the signaling of leptospiral LPS *via* TLR2 [24]. Stimulation of RAW264.7 macrophages in serum-free medium, supplemented with increasing doses of recombinant soluble CD14 (sCD14) revealed that both RANTES and NO productions, as well as IFN-β mRNA production in response to the shorter M895 LPS were dependent on the presence of sCD14 (Fig 4A and 4B). Surprisingly, the presence of sCD14, even at 500 ng/mL, did not trigger the production of NO by the full-length L495 leptospiral LPS (Fig 4A). sCD14 mediates TLR4-signaling by binding and delivering LPS to the TLR4-MD2 complex [34]. Thus, to further characterize the role of sCD14 in the M895 LPS signaling, binding assays were performed on high resolution 20% acrylamide gels

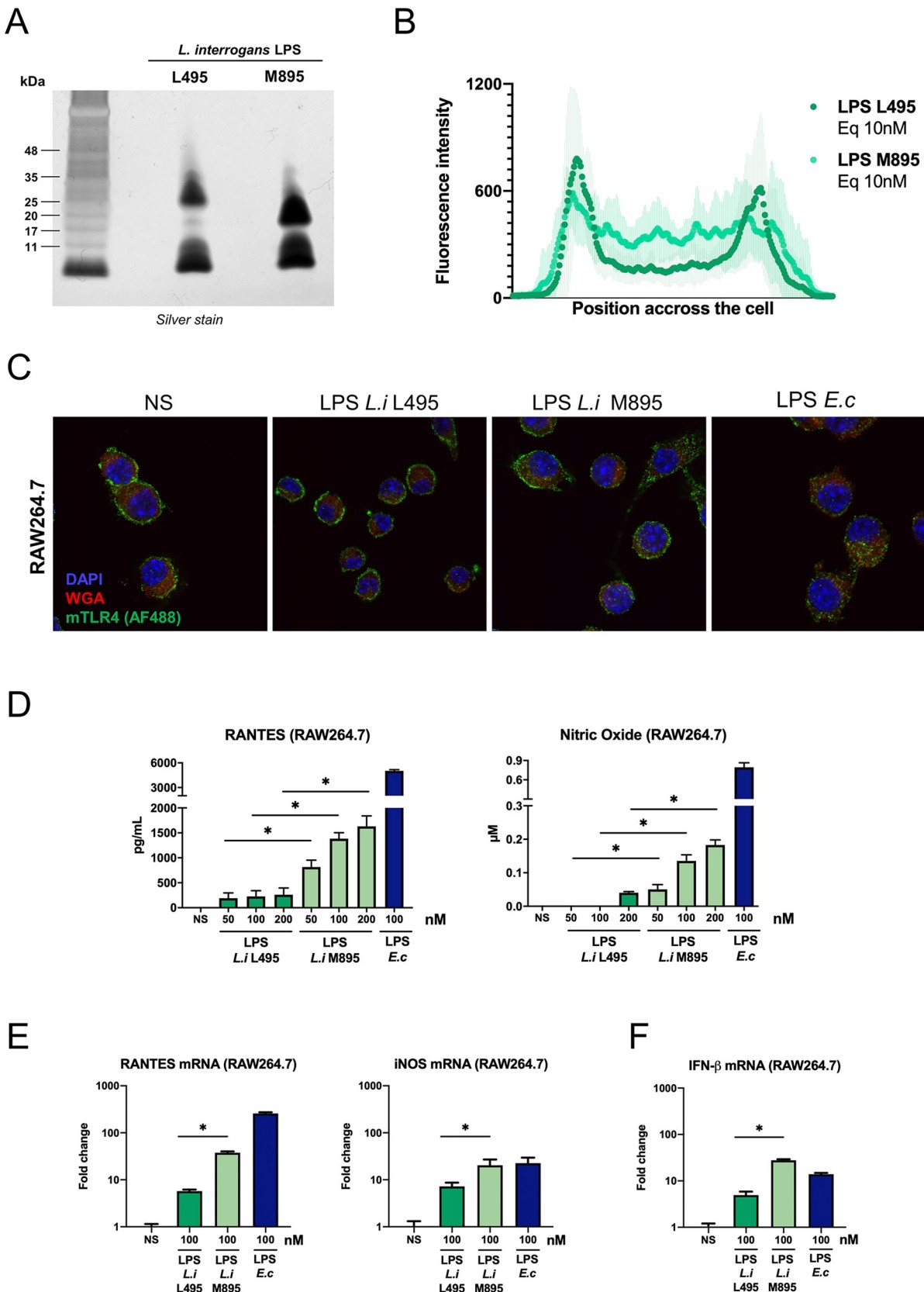

**Fig 3. Shorter leptospiral LPS induces more TLR4-TRIF responses. A)** Silver staining analyses of two LPS: the full-length LPS of the WT L495 strain and the shorter LPS of the mutant M895 strain after SDS-PAGE with the equivalent of 0.1 pmol of LPS/well. Positions of standard molecular mass markers are shown on the left. **B)** Average mouse-TLR4 (mTLR4) fluorescence profiles quantified on cross-section of RAW264.7 cells either stimulated by the equivalent 10 nM ($\approx$ 100 ng/mL) of the full-length LPS L495 (dark green) or the shorter LPS M895 (light green). The two LPS used were purified from *L. interrogans*, serovar Manilae WT strain L495 or mutant strain M895. Data are represented as mean of $>$ 40 cells and shades correspond to +/- SD. **C)** Representative images of confocal IF analyses of RAW264.7 cells upon 1h stimulation with the equivalent of 10 nM of LPS L495, LPS M895 or LPS of *E. coli*, and staining for mTLR4, glycoproteins (Weat Germ Agglutinin, WGA) and nuclei (DAPI). **D)** Production of RANTES and NO by RAW264.7 cells after 24h stimulation with 50-200 nM ($\approx$ 0.5-2 µg/mL) of full-length LPS of *L. interrogans* L495 (dark green), shorter LPS of *L. interrogans* M895 (light green) or LPS of *E. coli* (blue). **E)** mRNA levels of RANTES, iNOS and **F)** IFN-β in RAW264.7 cells after 24h stimulation with 100 nM of full-length LPS of L495 (dark green), shorter LPS of M895 (light green) or LPS of *E. coli* (blue). Data are represented as mean (+/- SD) of $n$ = 3/4 technical replicates and are representative of at least 3 experiments. Statistical analyses were performed using the non-parametric Mann-Whitney test.

and visualized by silver staining and Western Blots (WB). On silver staining, the band corresponding to sCD14 disappeared in the presence of both LPS (that remains in the upper part of the gel), suggesting a binding between the molecules. However, the disappearance of the sCD14 band was much more prominent with 0.01 pmol of the shorter M895 LPS than with the same dose of the L495 LPS (Fig 4C, left panel), suggesting a better binding of sCD14 to the short M895 LPS. These results were confirmed by WB targeting sCD14 *via* the polyHistidine tag (Fig 4C, right panel), on which the shifted sCD14 upper band was detectable and much more visible in the presence of the short M895 LPS. Overall, these results suggest that the full-length LPS has a lower affinity for sCD14 than the shorter LPS, which could explain why it escapes mTLR4 internalization and TRIF-responses.

## Co-purifying lipoproteins contribute to the escape of internalization

Because of the striking lack of TLR2 activity of the saprophytic Patoc LPS (S3B Fig and S3C Fig), we also investigated whether lipoproteins co-purifying with the leptospiral LPS could participate in the escape of TLR4-TRIF pathway. Indeed, one of the particularities of *L. interrogans* LPS extracted in the phenolic phase using the Westphal method is that it always co-purifies with lipoproteins [35]. These contaminants are responsible for the TLR2 activity of the leptospiral LPS, but the role of these co-purifying lipoproteins has never been further investigated. We therefore repurified LPS preparations of the Verdun strain. Quality controls of this repurification on HEK reporter system (Fig 5A) indicated that the TLR2 activity was almost totally removed in the repurified LPS (compared to the original), without altering mTLR4 activity. Stimulation of TLR2 KO BMDMs confirmed that, compared to the original LPS, there was no contribution of TLR2 to the signaling of the repurified LPS (Fig 5B). Furthermore, silver staining analysis showed that the overall structure of the LPS was not altered by the treatment (Fig 5C). Among other lipoproteins, we identified the LipL21 (Fig 5C) as co-purifying with the LPS; and it was significantly decreased upon repurification. Stimulation of RAW264.7 cells by the original and repurified LPS showed that the localization of mTLR4 was slightly modified when stimulated with the repurified LPS compared with the original LPS (Fig 5D and 5E). In BMDMs we observed consistently increased production of TRIF-dependent RANTES and NO, but not of MyD88-dependent KC upon stimulation with the repurified LPS (Fig 5F). These findings were confirmed by mRNA analyses of RANTES and iNOS (Fig 5G). Finally, IFN-β mRNA was also increased after stimulation by the repurified LPS (Fig 5H). These results suggest that the presence of co-puryfing lipoproteins would have an inhibitory role and would impair the leptospiral signaling through TRIF.

To confirm the implication of the co-purifying lipoproteins in escape from the mTLR4-TRIF pathway, we also treated the leptospiral LPS preparations with either proteinase K or lipase. Results with the HEK reporter system showed that these treatments did not affect

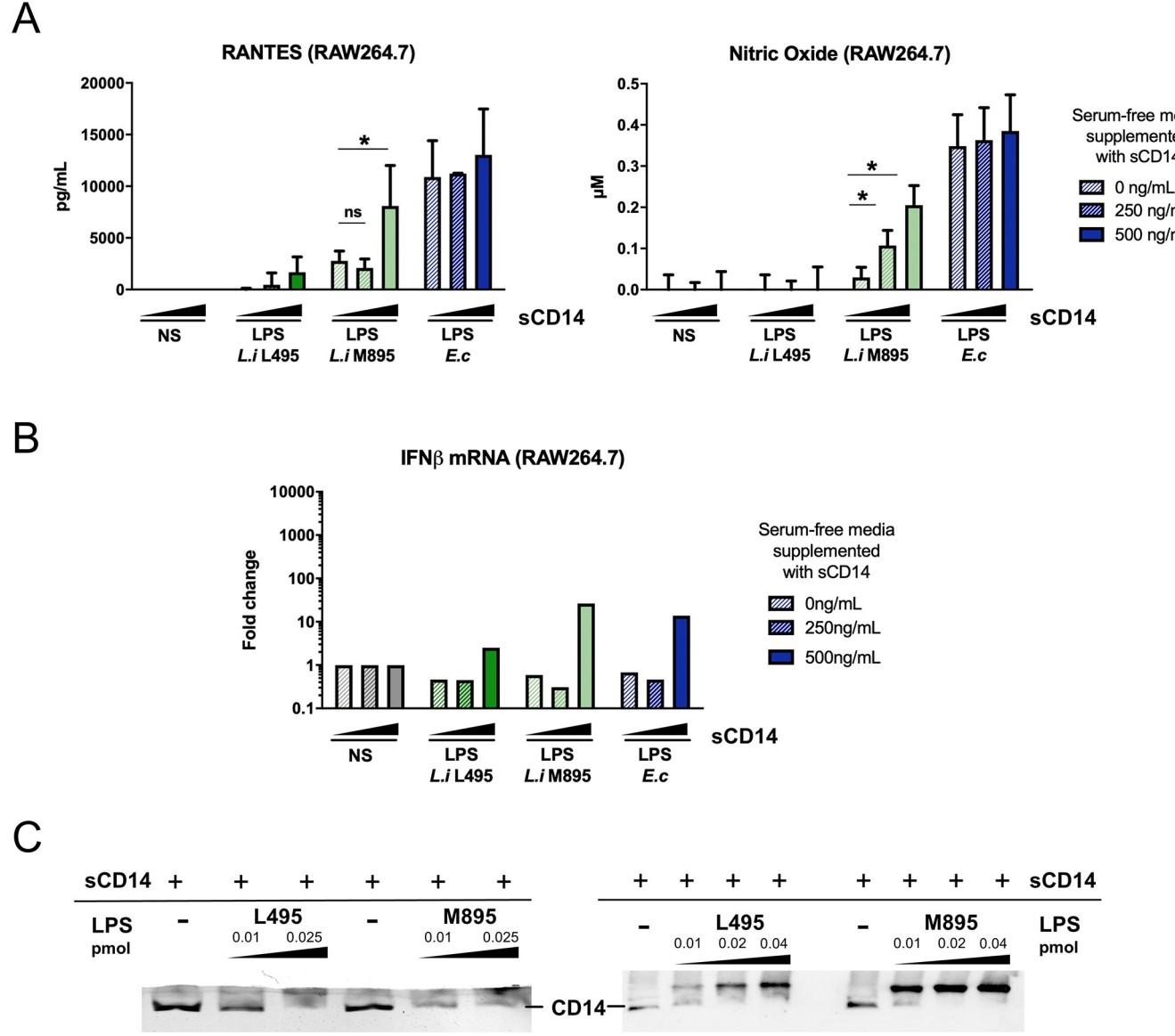

**Fig 4. Soluble CD14 participates in the enhanced signaling of the shorter leptospiral LPS. A)** Production of RANTES and NO by RAW264.7 cells after 24h stimulation in serum free conditions, with 0-500 ng/mL of recombinant bovine sCD14 and after stimulation with 100 nM ($\approx$ 1 $\mu$g/mL) of full-length LPS of *L. interrogans* L495 (dark green), shorter LPS of *L. interrogans* M895 (light green), or 1 $\mu$g/mL of LPS *E. coli* (blue). **B)** mRNA levels of IFN-$\beta$ in RAW264.7 cells after 24h stimulation with 0-500 ng/mL of recombinant bovine sCD14 and after stimulation with 100 nM ($\approx$ 1 $\mu$g/mL) of full-length LPS of *L. interrogans* L495 (dark green), shorter LPS of *L. interrogans* M895 (light green), or 1 $\mu$g/mL of LPS *E. coli* (blue). **C)** Binding assay of recombinant bovine sCD14 to the full-length L495 and shorter M895 leptospiral LPS. Native 20% gel electrophoresis of the LPS (0.01–0.04 pmol/well) + sCD14 complexes revealed by silver staining (left panel, 100 ng/well of sCD14) or Western Blot targeting sCD14-His (right panel, 200 ng/well of sCD14) show the band of sCD14 that disappears or gets shifted in presence of LPS. Data are represented as mean (+/- SD) of *n* = 3/4 technical replicates and are representative of at least 3 independent experiments. Statistical analyses were performed using the non-parametric Mann-Whitney test. RT-qPCR performed in technical duplicates were not used in the statistical analyses.

mTLR4 activity (S5A Fig and S5B Fig). Interestingly, lipase digestion only (not proteinase K) resulted in the expected reduction in the TLR2 activity of the LPS (S5A Fig and S5B Fig). Consistent with the repurification data, leptospiral LPS preparations digested with proteinase K (and not lipase) induced increased RANTES production in macrophages (S5C Fig and S5D

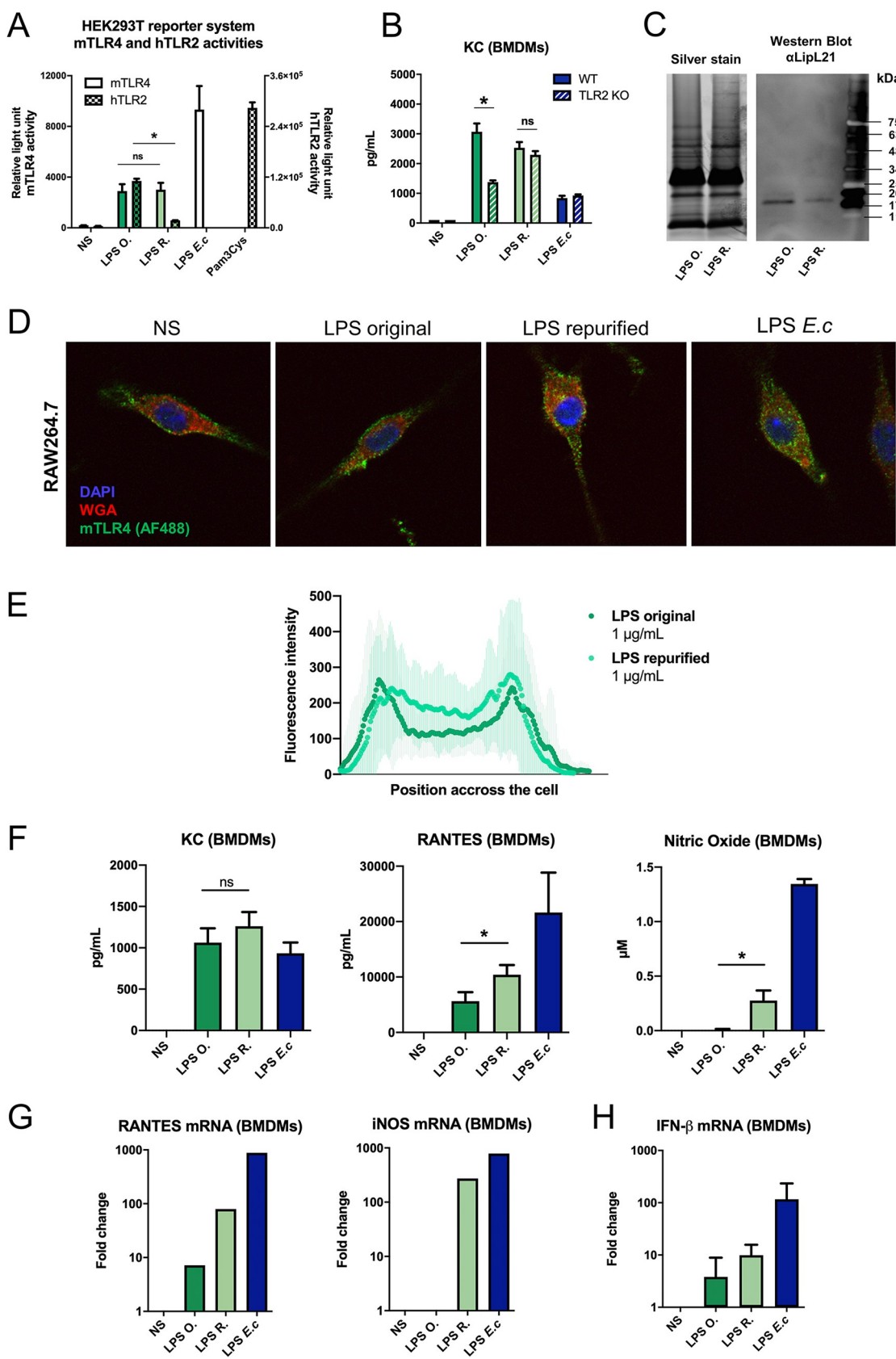

**Fig 5. Copurifying lipoproteins contribute to the escape of TLR4 internalization. A)** Human embryonic kidney cells (HEK293T) reporter system for mouse-TLR4 (left axis) and human-TLR2 (right axis) activities of 1 μg/mL of the original LPS of *L. interrogans* strain Verdun before repurification (LPS O., dark green) or repurified leptospiral LPS (LPS R., light green), with the corresponding controls: 100 ng/mL of LPS of *E. coli* for mTLR4 activity and 100 ng/mL of Pam3Cys for hTLR2 activity. **B)** Production of KC by WT and TLR2 KO BMDMs after 24h stimulation with 1 μg/mL of LPS O. (dark green), LPS R. (light green) or 100 ng/mL of LPS of *E. coli* (blue). **C)** Silver staining analyses of original or repurified LPS of *L. interrogans* after SDS-PAGE with the equivalent of 1 μg of LPS/well (left panel) and Western Blot showing LipL21 contamination in the two LPS preparations (right panel). Positions of standard molecular mass markers are shown on the right. **D)** Representative images of confocal IF analyses of RAW264.7 macrophage-like cells upon 1h stimulation with 100 ng/mL of LPS of *Leptospira interrogans* original, repurified, or LPS of *E. coli* and staining for mouse-TLR4 (mTLR4), glycoproteins (Weat Germ Agglutinin, WGA) and nuclei (DAPI). **E)** Average mTLR4 fluorescence profiles quantified on cross-section of the cells either stimulated by the original (dark green) or the repurified leptospiral LPS (light green). Data are represented as mean of > 30 cells and shades correspond to +/- SD. **F)** Production of KC, RANTES and NO by BMDMs after 24h stimulation with 1 μg/mL of LPS O. (dark green), LPS R. (light green) or LPS of *E. coli* (blue). **G)** mRNA levels of RANTES, iNOS and **H)** IFN-β in BMDMs cells after 24h stimulation with 1μ/mL of original (dark green) or repurified (light green) LPS of *L. interrogans* or LPS of *E. coli* (blue). Data are represented as mean (+/- SD) of *n* = 3/4 technical replicates and are representative of at least 3 independent experiments. Statistical analyses were performed using the non-parametric Mann-Whitney test. RT-qPCR performed in technical duplicates were not used for statistical analyses.

Fig). We checked that the proteinase K treatment did not alter the structure of the leptospiral LPS (S5E Fig) and that it efficiently degraded flagellin (S5F Fig). All together, these data suggest that the lipoproteins contribute to the TLR4-TRIF escape *via* the protein moiety and not *via* the lipid anchors. LipL32 was also found to copurify with the leptospiral LPS, and to be decreased upon repurification (S5G Fig). Since it is the major lipoprotein present in the outer membrane of leptospires and is the only one to have been shown to be a TLR2 agonist [24,28], we also purified and stimulated macrophages with the LPS of *L. interrogans* Manilae mutant strain M933 ΔLipL32. Although the molar concentration of the LPS preparations was not measured, using similar amounts (as estimated by weight), the mutant LPS did not have a lower TLR2 activity than the WT LPS (S5H Fig). Additionally, there was no increased production of RANTES (S5I Fig). These results suggest that the phenotype was not due to a single specific lipoprotein. This led us hypothesize that lipoproteins are redundant regarding escape of the internalization of TLR4.

## Whole bacteria induce responses consistent with LPS signaling properties

Our data showed that leptospiral LPS escapes mTLR4 internalization and that full-length LPS and co-purifying lipoproteins are important features for avoiding TRIF responses. Therefore, we investigated whether the LPS responses could mimic an infection with whole bacteria. No internalization of mTLR4 was visible 1-hour post infection of macrophages with live WT *L. interrogans* Manilae L495, whereas infection with live Manilae M895 induced internalization of the mTLR4 (Fig 6A and Fig 6B). Furthermore, we tested the TRIF-dependent RANTES and NO productions 24 h post infection and we observed that, although there was no significant difference in RANTES production, the WT L495 strain induced less NO than the mutant M895 strain (Fig 6C). Of note, experiments with live bacteria were performed using MOI comprised between 10 and 50, which could relate to 1 μg/mL of LPS according to our estimations, although such comparison must be interpreted with caution. These results show that the signaling of the leptospiral LPS is, to some extent, representative of the responses induced by whole bacteria, and that live WT leptospires also escape the mTLR4 internalization and part of the TRIF-dependent antimicrobial responses.

## Discussion

Overall, our data show that the leptospiral LPS escapes mTLR4 internalization and consequently avoids the activation of TRIF-dependent responses such as NO, IFNβ and RANTES in

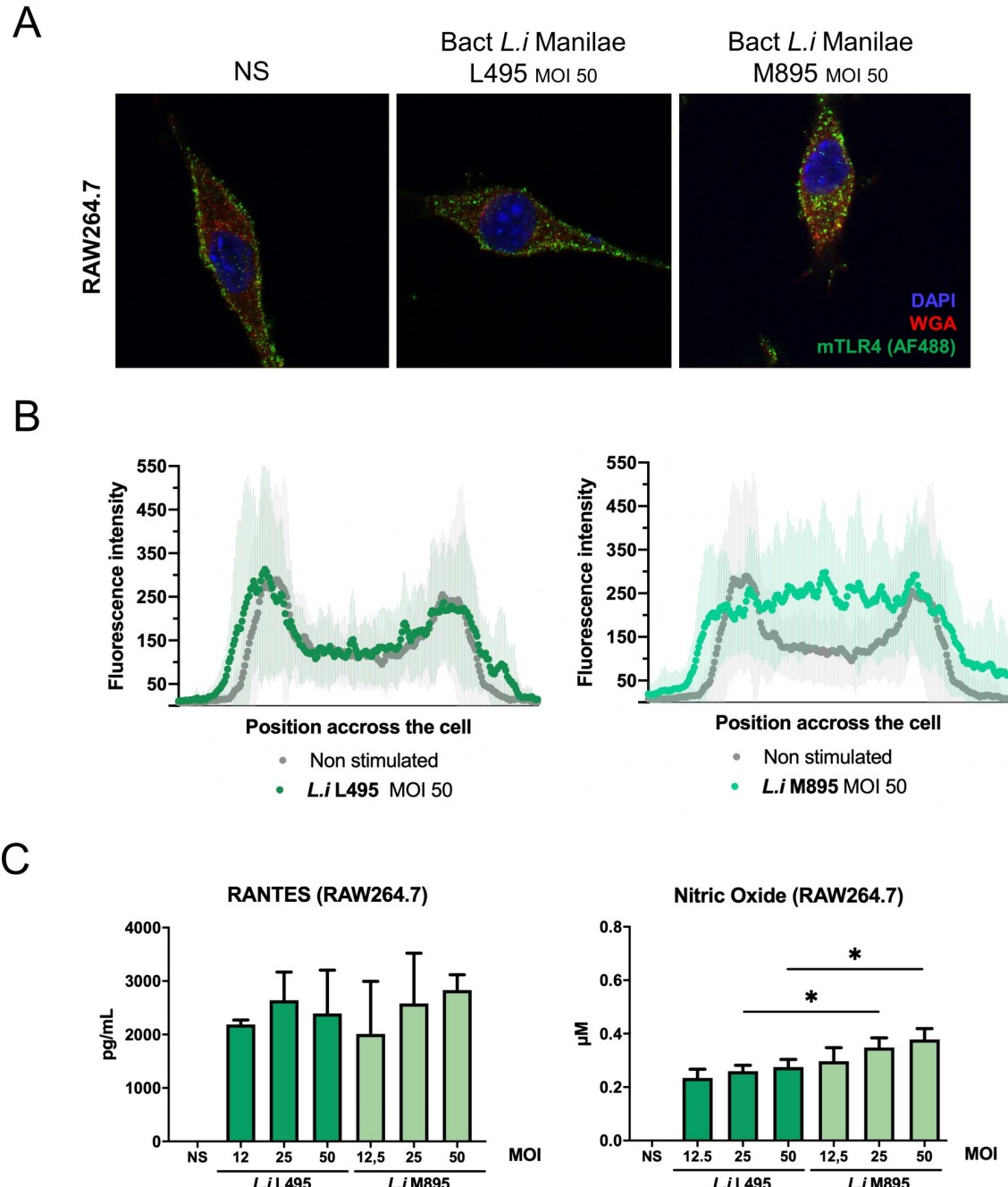

**Fig 6. Whole bacteria induce responses similar to corresponding LPS. A)** Representative images of confocal IF analyses of RAW264.7 macrophage-like cells upon 1h stimulation with either MOI 50 of whole live bacteria *L. interrogans* L495 or mutant M895 and staining for mouse-TLR4 (mTLR4), glycoproteins (Weat Germ Agglutinin, WGA) and nuclei (DAPI). **B)** Average mTLR4 fluorescence profiles quantified on cross-section of the cells ($n > 30$) either non-stimulated (grey) or stimulated by whole *L. interrogans* L495 or mutant M895 (green). **C)** Production of RANTES and NO by RAW264.7 cells stimulated 24h by MOI 12.5–50 of whole live bacteria *L. interrogans* WT L495 (dark green) or mutant M895 (light green). Data are represented as mean (+/- SD) of $n = 3/4$ technical replicates and are representative of at least 3 independent experiments. Statistical analyses were performed using the non-parametric Mann-Whitney test.

murine macrophages. Using shorter and repurified LPS, we demonstrated the effect to be dependent on the presence of a full-length O antigen and co-purifying lipoproteins. Furthermore, we showed that it is the protein moiety of these lipoproteins that is important for the

escape of mTLR4 internalization. In addition, we showed that the full-length O antigen of the leptospiral LPS impairs the binding to CD14, known to be instrumental for TLR4 endocytosis. Finally, we showed that the phenotype described with leptospiral LPS was representative of an infection by whole live bacteria. These data therefore suggest that the leptospiral LPS, that was already known to fully escape hTLR4 recognition in humans that are sensitive, also escapes part of the TLR4 responses in mice. We therefore hypothesize that this partial escape of recognition by the innate immune system could play a key role in the stealthiness of the bacteria and chronic kidney colonization in the resistant mouse model (Fig 7).

Interestingly, CD14 was described as essential for TLR4 endocytosis and TRIF pathway activation, at both high and low LPS doses and in both macrophages and endothelial cells [36,37]. Our results also suggest that sCD14 enhances TRIF responses to the shorter LPS of the leptospiral mutant M895. This could be due to a stronger binding of the mutant LPS to sCD14 compared to the full-length LPS. Conversely, we hypothesize that the poor binding of the leptospiral LPS to sCD14 could limit TLR4-TRIF signaling. Furthermore, we showed that at

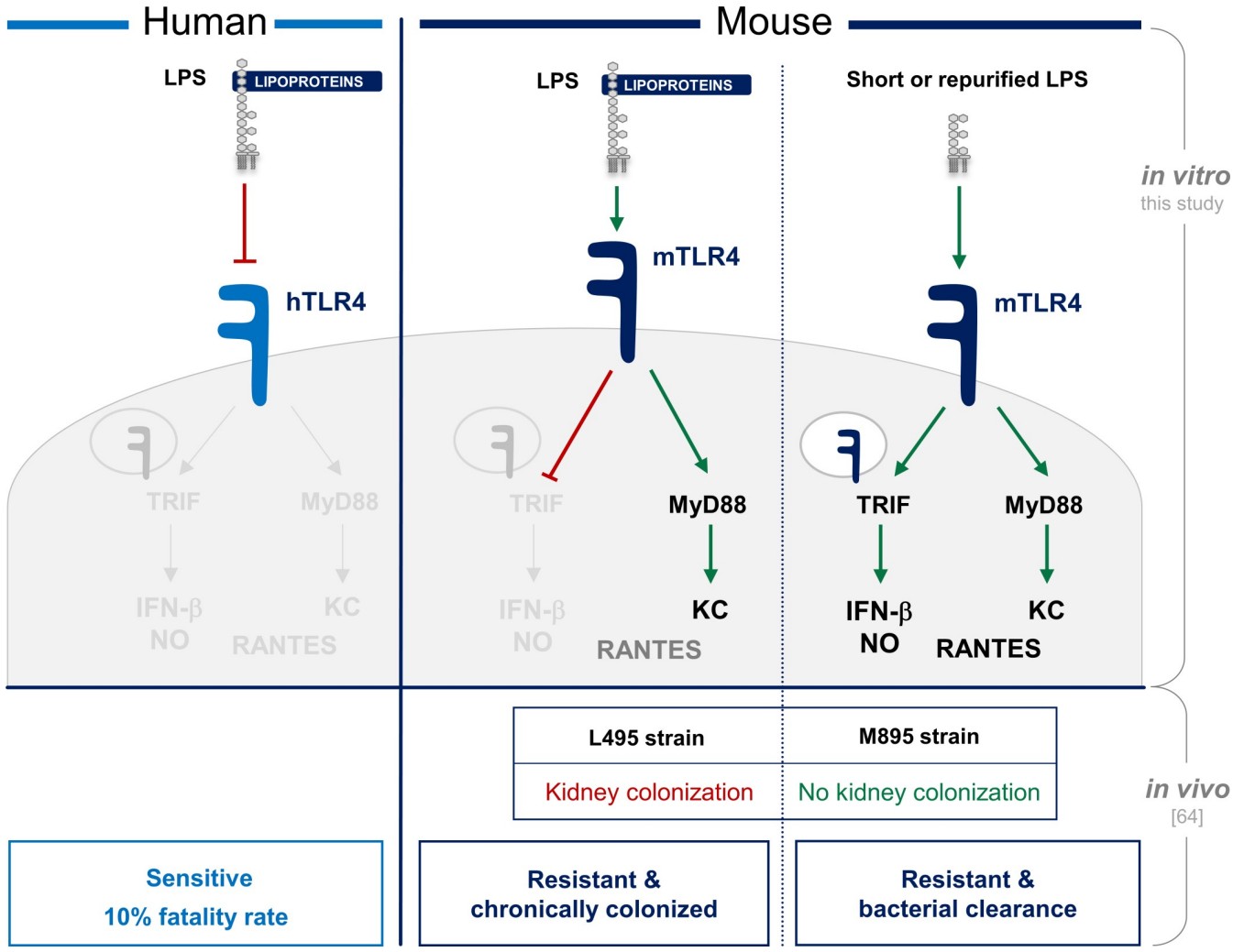

**Fig 7. Summary of the TLR4 recognition of the leptospiral LPS in human and mouse.** Schematic representation of the difference between: the lack of TLR4 recognition in humans that leads to full immune escape of the leptospires from TLR4 and may cause susceptibility to acute leptospirosis; and the partial lack of TLR4 recognition in mice that (although protects from acute leptospirosis) could account for partial immune escape of the leptospires and contribute to the chronicity of the infection in mice.

lower concentrations, leptospiral LPS escapes both the MyD88 and the TRIF pathways, as illustrated by the lack of KC and NO production. Again, the poor binding of the LPS to CD14 is consistent with these results, given that CD14 is essential for both the MyD88 and the TRIF pathways upon stimulation by low concentrations of LPS [20,36–41]. This is reminiscent of another study that described poor binding of CD14 to atypical LPS of *Helicobacter pylori* and *Porphyromonas gingivalis* [42]. Given that full-length and shorter LPS of *L. interrogans* Manilae strains L495 and M895 have the same lipid A, it is reasonable to hypothesize that CD14 poorly binds to the leptospiral LPS because of steric hindrance due to the length of the O antigen, which could be a cause of its altered signaling. This would be consistent with previous studies showing that smooth LPS (S-LPS) depends on CD14 to activate cells [40,43]. On the other hand, it is established that rough LPS (R-LPS) depends on CD14 mainly for TLR4 endocytosis [21,22], although some studies reported both *in vitro* and *in vivo* that R-LPS could benefit from the presence of CD14 [21,44]. Accordingly, several other studies have described R-LPS as a more potent agonist of the innate immune system than S-LPS [40,44–47]. Overall, such findings are consistent with our results showing that the shorter LPS naturally signals more *via* TRIF than the full-length LPS, and that sCD14 contributes to this enhanced signaling. Complementation experiments restoring the complete structure of the shorter LPS mutant would have been valuable to confirm a direct link between the LPS structure and signaling properties. However, the available genetics tools for *L. interrogans* are limited, hence making complementation a real challenge. Previous attempts at complementing the M895 mutant using a modified transposon have failed as none of the selected transformant strains exhibited a full-length LPS, almost certainly due to the very large transcriptional units that make up the LPS biosynthetis locus [7].

In addition, we showed that the copurifying and omnipresent lipoproteins were also responsible, at least in part, for the lack of TLR4 intracellular signaling. We hypothesize that these lipoproteins also affect CD14 binding and that they could potentially play a role in CD14 sequestration. Indeed, CD14 was shown to be important for the signaling of triacylated lipoproteins *via* TLR2-TLR1 [48] especially in the case of spirochetal lipoproteins [24,49–52]. Furthermore, we hypothesize that this phenomenon might not be specific to *Leptospira* and could be conserved among bacteria. Indeed, there are other LPS that possess the ability to activate TLR2, such as *H. pylori* or *Porphyromonas* spp. [53–55]. Many studies identified lipoproteins as the TLR2-activating contaminants in LPS preparations [55,56]. In the case of *L. interrogans*, LipL32, the most abundant lipoprotein [27], was shown to be a TLR2 agonist [24,28,57]. Furthermore, many groups worked on purifying these contaminated LPS preparations and reported the loss of TLR2 activity upon extensive repurification [55,58]. Surprisingly, the loss of TLR2 activity upon repurification did not necessarily blunt the stimulatory properties of the LPS. Although not discussed by the authors, several studies showed similar or enhanced TNF-α production upon stimulation by repurified LPS [55,58]. Such findings are consistent with our results that showed enhanced TRIF signaling upon repurification and removal of the copurifying lipoproteins. It is of note that the efficiency of these repurification processes differs according to LPS. LPS naturally harboring TLR2 activity (*H. pylori*, *P. gingivalis*, *L. interrogans*) seem to be more difficult to repurify than classical LPS from *Salmonella* [24,55,58]. Purification of the leptospiral LPS from the phenol (and not aqueous) phase of the Westphal method [35], most probably due to the hydrophobicity of its O chains, could contribute to the co-purification with other hydrophobic molecules such as lipoproteins, hence making this LPS harder to purify with the classically described methods. Further biochemical studies would be required to understand whether those lipoproteins may directly impair the LPS binding to CD14 and how they may promote the TLR2 signaling while dampening TLR4-TRIF responses. One interesting result of our study was obtained with the LPS from the mutant M933 ΔLipL32

that showed no enhanced TRIF activation compared with the WT LPS, suggesting that the lipoproteins could be redundant. In the case of *L. interrogans*, over 170 putative lipoproteins have been described, but their function remains mostly undetermined. We showed that LipL21 contributes to the escape of the immune system *via* very tight binding to peptidoglycan (PG) that required 4 hours boiling in SDS to be detached from the PG [30], hence preventing its recognition by the PRRs of the NOD family NOD1 and NOD2 [30]. Therefore, *L. interrogans* in which the abundance and variety of the lipoproteins is unusual, allowed us to highlight once more a potential new role for bacterial lipoproteins in innate immune escape. In particular for this phenotype, the redundancy of lipoproteins highlights this mechanism as a potential key strategy employed by leptospires to evade the immune response.

Although we mainly focused our study on the potential role of the lipoproteins and the O antigen that could impair the binding to CD14, we cannot exclude the possibility that other host factors might be involved. Among them is the lipopolysaccharide-binding protein (LBP). The LBP functions as a catalyst allowing for the binding of monomeric LPS to CD14 that is then transferred to TLR4 *via* interaction with the myeloid differentiation factor 2 (MD2) [34]. Although the LBP greatly facilitates the transfer of the LPS to the soluble CD14, it is not essential to the signaling [59]. Just like CD14, LBP plays an important role in the CD14-dependent endocytosis of TLR4 [19] and plays a crucial role in the signaling of low doses of LPS *via* MyD88 [60]. Interestingly, in the case of *L. interrogans*, LBP was shown to be the only serum factor upregulated in the blood of patients upon infection [61]. Therefore, we cannot exclude that LBP contributes to the effect. Finally, we cannot exclude the possibility that the escape of TLR4 internalization upon stimulation by leptospiral LPS could be due to a CD14/LBP-independent mechanism. For instance, stimulation by a mycobacterial phenolic glycolipid (PGL) induces similar dampening of the TRIF-associated responses in macrophages [62]. This effect was associated with TRIF protein level (and not mRNA) decrease after at least 6h of stimulation by PGL [62]. However, we do not favor this hypothesis, given that we consider the lack of TRIF-response to be the direct consequence of the lack of TLR4 internalization that we showed to be already apparent at 1 h post stimulation.

Interestingly, the M895 mutant of Manilae with shorter LPS is attenuated in hamsters [7] and has lost its ability to colonize kidneys in BALB/c mice [63]. Other examples of such a correlation between the reduced size of the LPS and an altered infection course of the corresponding bacteria have been published. For example, the deep-rough LPS mutant of *B. bronchiseptica* is defective for lung colonization and pulmonary infection in a murine model [47]. Together, our work suggests the *in vivo* relevance of the role of the O antigen of leptospiral LPS for TLR4 signaling and confirms its importance to escape part of the innate immune responses, potentially favoring kidney colonization in mice. Among the numerous factors that could contribute to the attenuation of the leptospiral mutant, one could hypothesize that the stronger binding of the mutant LPS to CD14 and subsequent enhanced signaling could benefit the infected host in the fight against the pathogen. In the case of the M895 mutant, we have shown that it induces more NO than the WT strain, and we hypothesize that the activation of the TLR4-TRIF pathway could, in addition to the activation of TLR2, contribute to a strong innate immune recognition by the infected host, leading to better clearance and no kidney colonization in mice (Fig 7). Our results thus provide for the first time a plausible explanation for the basis of attenuation of the LPS mutant M895. Accordingly, it was recently shown that *L. interrogans* serovar Autumnalis strain 56606v, which possesses an LPS atypically devoid of TLR2 activity and signaling only through TLR4 [64], leads to a self-resolving leptospirosis and does not induce kidney colonization in mice. To our knowledge, besides the Patoc LPS described here, this is the first evidence of a leptospiral LPS naturally devoid of TLR2 activity. Upon infection with that specific leptospiral strain, TLR4 was shown to be essential to the

resolution of the infection, and authors described a TLR4-dependent iNOS activation that they hypothesize could contribute to the leptospiral clearance in mice [64]. Similarly, we show that LPS of *L. biflexa* Patoc, that is also devoid of TLR2 activity, induces TLR4-TRIF responses and we hypothesize that such mechanism contributes to the rapid clearance of the saprophytic bacteria by infected hosts [5]. Conversely, these findings suggest that full length O antigen of the WT LPS and co-purifying lipoproteins contribute to the dampening of the TRIF signaling and potentially participate in the stealthiness of the pathogenic leptospires, leading to the productive infection and colonization of the kidneys in mice. Consistent, the lack of the gene encoding iNOS protein in mice has negligible effect on the outcome of infection by *L. interrogans* [65,66]. Moreover, to our knowledge, only one study described a protective role of the TRIF adaptor upon induction of severe leptospirosis [67] induced with 2 x $10^8$ leptospires per mouse. Such findings are consistent with our results showing that leptospires at high MOI, unlike LPS, can trigger NO in macrophages at 24h post infection although no mTLR4 internalization was visible at 1h post infection. This suggests that TRIF activation is impaired, but not entirely abolished, upon infection. Moreover, we cannot exclude the possibility that TRIF could be induced later, or by TLR2 activation [68–73] by lipoproteins upon infection, as illustrated by the *Leptospira*-induced production of NO in C3H/HeJ TLR4-defective mice [73]. Overall, these studies suggest that TRIF plays a moderate role in the outcome of the infection by WT bacteria, unlike MyD88 [74], and could be key to determine the outcome of both the acute phase and chronic kidney colonization in mice. Clinical studies have consistently reported that the escape of the TRIF-dependent RANTES production was important in clinical leptospirosis [75]. Furthermore, it was also reported that anti-leptospiral IgM levels were slightly reduced in Trif*Lps2* mice [67]. The low TRIF activation that we described here could therefore have impacts beyond the first line of innate immune defense and could also potentially dampen the humoral responses against *L. interrogans*.

To conclude, the recognition by mTLR4 is essential for host resistance against leptospirosis. Although leptospiral LPS is recognized by the mTLR4, we described here that *in vitro* it escapes the TRIF arm of the TLR4 response, hence dodging antimicrobial responses. Accordingly, another study [64] has demonstrated that such mechanism is relevant *in vivo*. The novelty of the present work resides in the study of the signaling properties of the leptospiral LPS in its "crude" form, and investigation of the role played by the physiologically relevant copurifying lipoproteins. We propose that this partial innate immune escape of the leptospiral LPS in mice could be key to the stealthiness of the bacteria and to the chronicity of leptospires in resistant mice.

## Materials and methods

### *Leptospira* strains and cultures

*L. interrogans* serovar Icterohaemorrhagie strain Verdun; serovar Manilae strain L495, strain M895 (shorter LPS [7]) and strain M933 (ΔLipL32 [76]); serovar Copenhageni strain Fiocruz L1-130, and *L. biflexa*: serovac Patoc strain Patoc were used in this study, and grown in liquid Ellinghausen-McCullough-Johnson-Harris medium (EMJH) at 30˚C without antibiotics and agitation. All strains were subcultured weekly at 5 x $10^6$ leptospires/mL (or twice a week in the case of *L. biflexa*) and hence kept in exponential phase of growth.

### Leptospiral LPS purification and quantification

LPS was extracted by the classical hot phenol/water protocol adapted from Westphal and collaborators [35] and described recently [77]. Silver staining analyses were performed as described by Tsai and Frasch [78] and reviewed recently [77]. LPS preparations were

quantified by EndoQuant technology (Dijon, France) (direct assay to quantify the 3-hydroxyl-ated fatty acids linked to the disaccharide of the lipid A). Lipoprotein contamination was assessed by SDS-PAGE and Western Blot (WB). Nitrocellulose membranes (BioRad) were blocked with 5% w/V BSA in Tris-Buffer-Saline-Tween (TBS-T) for 1 h at room temperature (RT), before staining over night at 4˚C with 1:10 000 anti-LipL21 or anti-LipL32 polyclonal rabbit antiserum (kindly provided by Dr D. Haake) described previously [79]. Membranes were washed 3 times with TBS-T and stained with 1:10 000 HRP-coupled anti-rabbit (anti-Rabbit IgG, HRP linked antibody from donkey, GE HealthCare) in blocking buffer for 1 h at RT. Development was performed after 3 washes in TBS-T and using Clarity ECL reagents (BioRad) and automatic exposure set up on ChemiDoc imaging system (BioRad).

## Additional purification of LPS

LPS was further purified to remove the TLR2 activating co-purifying molecules. LPS was re-extracted by the isobutyric acid/1M ammonium hydroxide method [80] and further purified by enzymatic treatment to remove DNA, RNA and proteins, then extracted with a mixture of solvents to remove phospholipids and lipoproteins as described [81]. LPS preparations were also treated with proteinase K (from *Tritirachium album*, Qiagen) or lipase (from *Burkholderia* ssp., Sigma, active on triacylated lipoproteins) at final concentrations varying from 1 X to 4 X the amount of LPS. All treatments were performed at 37˚C for 3 h.

## LPS and CD14 binding assays

LPS-CD14 binding assays were performed by incubating 100 ng/mL of recombinant bovine sCD14-His [82] with various concentrations of leptospiral LPS in PBS for 1 h at 37˚C and under slow agitation at 300 rpm. Samples were then loaded on polyacrylamide native gels (stacking 4.5%, running 20%), ran in TG native buffer (BioRad) at constant voltage of 50 V for 6-8 h on ice and used for WB or silver staining. Silver staining was performed using the Pierce coloration kit (ThermoFisher) according to the manufacturer's instructions. Images were acquired within 30 min of development. WBs were performed after turbo transfer to nitrocellulose membrane (BioRad) and blocking in 5% w/V BSA in PBS-Tween (PBS-T). Membrane was then stained with 1 μg/mL of HRP coupled anti-His (monoclonal mouse anti-polyHistidine-Peroxidase, Sigma-Aldrich) in 1% w/V BSA in PBS-T over night at 4˚C. Signal was visualized after three washes and with Clarity ECL reagents (BioRad) using automatic exposure on ChemiDoc imaging system (BioRad).

## Human embryonic kidney cells reporter system

Human embryonic kidney cells (HEK293T) cells were cultured in complete DMEM composed of DMEM GlutaMAX (Gibco), supplemented with 10% V/V heat inactivated Fetal Calf Serum (Hi FCS, Gibco), 1 mM sodium pyruvate (Gibco) and 1 X non-essential amino acids (Gibco). Transfections were performed in 24-well plates (TPP) at 60% confluence with a final amount of 300 ng of DNA per well (50 ng of NF-κB reporter luciferase; 25 ng of internal control β-galactosidase; 50 ng of hTLR2; completed with empty pcDNA3.1 vector) and using FuGENE HD 1x (Promega) in serum-free conditions following the supplier's indications. Briefly, DNA and transfection reagents were incubated for 25 min at RT before addition to the cells. The cells used for TLR4 activity were stably transfected with mTLR4, mCD14 and mMD2 (Invivogen). Cells were stimulated 24 h post transfection for 6 h with LPS, and controls for TLR4 or TLR2 activity, namely LPS from *E. coli* O111:B4 UltraPure (Invivogen) or Pam3Cys (Invivogen) respectively. Cells were lysed in 100 μL of lysis buffer (25 mM Tris pH 8; 8 mM $MgCl_2$; 1% V/V triton X-100; 15% V/V glycerol; 1 mM DTT). 10 μL were used for β-galactosidase

assay by addition of 100 μL of substrate (1 mg/mL ONPG in 0,06 M $Na_2HPO_4$; 0,04 M $NaH_2PO_4$; 0,01 M KCl; 0,001 M $MgSO_4$; 1 mM DTT), incubation at 37˚C for 20 min, followed by absorbance measurement at 450 nM on Elx808 analyzer (BioTek).10 μL were used for NF-κB reporter luciferase assay by addition of 100 μL of 1 mM luciferin and 0.1 M ATP in lysis buffer and analysis on Berthold Centro X100 luminometer. Luciferase reporter assay values were individually normalized by β-galactosidase and reported as NF-κB activation.

## Macrophages culture and stimulation

Bone marrow derived macrophages (BMDMs) were obtained after euthanasia by cervical dislocation of C57BL/6 adult mice (more than 6 weeks-old female or male): WT (Janvier, Le Genest, France), and Trif*Lps2*, MyD88 KO, TLR2 KO, TLR4 KO, all bred by the Institut Pasteur animal facility and described previously [65]. After isolation of the femurs, head of the bones were cut, and the bone marrow was flushed out with a 22 G needle. Cells were centrifuged at 300 g for 7 min and erythrocytes were lysed for 10 min with Red Blood Cells Lysis Buffer (Sigma-Aldrich). After a second centrifugation, the bone marrow cells were counted with Trypan blue (Gibco) and plated in culture dishes (TPP) with 5 x $10^6$ cells/dish in 12 mL of complete RPMI (RPMIc): RPMI (Gibco) with 10% V/V Hi FCS (Gibco), 1 mM sodium pyruvate (Gibco) and 1 X non-essential amino acids (Gibco). RPMIc was supplemented with 1 X antibiotic-antimycotic (Gibco) and 10% V/V L929 cells supernatant to differentiate them to macrophages (obtained from confluent cultures and filtered through 0,22 μm). Cells were incubated at 37˚C, 5% $CO_2$ for 7 days (with addition of 3 mL of differentiation medium at day 3). BMDMs were recovered at day 7 by scraping in Cell Dissociation Buffer (Gibco) and centrifugation at 300 g for 7 minutes. BMDMs were plated 3 h before stimulation at a concentration of 1 x $10^6$ cells/mL in 96-well plates (TPP) with 200 μL per well.

RAW264.7 (macrophage-like cell line) cells were plated the night before stimulation at a concentration of 0,5 x $10^6$ cells/mL in 96-well plates with 200 μL per well in RPMIc. For the experiments performed without FCS and with sCD14, FCS was progressively removed and the recombinant soluble bovine CD14-His [82] was added before stimulation. RAW264.7 cells or BMDM were stimulated for 24 h with leptospiral LPS, whole bacteria or other agonists: LPS of *E. coli* O111:B4 (UltraPure or EB, Invivogen), Poly(I:C) HMW (Invivogen), Pam3Cys (Invivogen).

## ELISA and Griess reaction

For mouse-KC/CXCL1 and mouse-RANTES/CCL5 quantification, ELISA was performed on cell culture supernatants that were kept at -20˚C. ELISA DuoSet kits (R&D Systems) were used according to the manufacturer's instructions. The nitric oxide (NO) was quantified in the fresh supernatant by the Griess reaction.

## RT-qPCR

Frozen mouse cells were used for RNA extractions that were performed using RNeasy Mini kit (Qiagen) and according to the supplier's instructions. Reverse transcription was performed immediately after extraction and using SuperScript II Reverse Transcriptase (Invitrogen) according to manufacturer's instructions. cDNA obtained was quantified using NanoDrop (Thermo Fisher Scientific), frozen and kept at -20˚C for qPCR. Five μg of cDNA were used for RT-qPCR using Taqman Universal Master Mix II (Applied Biosystems). Primers and probes are listed in S1 Table. RT-qPCR was performed on a StepOnePlus Real Time PCR System (Applied Biosystems) with the standard protocol (2 h) and fold change was calculated with the $2^{-\Delta\Delta C\tau}$ method on the average of the technical duplicates.

## Immunofluorescence staining, confocal microscopy and image quantification

Cells were seeded at 0,2 x $10^6$ cells/mL in 24-well plates containing 12 mm round coverslips (#1.5 mm thickness, Electron Microscopy Science) with 1 mL per well and stimulated as described above. Cells were fixed 1 h post stimulation with 3% V/V *para*-formaldehyde for 10 min and washed 3 times with PBS. Cells were stained as follows: 1- blocking with 2 μg/mL FcBlock (anti mouse CD16/CD32, Invitrogen) for 1 h at RT in microscopy buffer (5% w/V BSA (Sigma), 0,06% w/V saponin (Sigma) in PBS); 2- incubation with 0.5 μg/mL primary anti mouse-TLR4 (recombinant monoclonal rabbit IgG, clone #1203B, R&D Systems) in micros-copy buffer for 1 h at RT; 3- three washes with wash buffer (1% w/V BSA in PBS); 4- incuba-tion with 4 μg/mL secondary goat anti rabbit conjugated with AlexaFluor(AF)488 (goat IgG H +L, Invitrogen) in microscopy buffer for 1 h at RT; 5- three washes in wash buffer; 6- incuba-tion with 1 μg/mL DAPI(Invitrogen) staining nuclei and 2 μg/mL Wheat Germ Agglutinin (Invitrogen) conjugated with AF594 staining all glycoproteins in permeabilized cells in 5% w/ V BSA in PBS for 10 min at RT; 7- three washes with PBS. Coverslips were mounted with Fluoromont (Sigma) and kept at RT over night before imaging. All images were obtained with a Leica SP5 confocal microscope using a 63x, 1.4 N.A., oil immersion objective. The following parameters were used: pinhole size of 1 Airy Unit (default setting), scanning frequency of 200 Hz, line average of 4. UV laser was used for DAPI, Argon for AF488 and 594HeNe for AF594, with power between 5 to 15%. Sequential acquisition by channel was performed using a hybrid detector for AF488 (gain range from 70% to 100%) and photomultipliers for AF594 and DAPI (voltage between 700 to 900 V). Raw images were analyzed with ImageJ-Fiji [83]. Contrast and brightness were homogeneously enhanced (10%) on the representative images shown for each experiment. mTLR4 fluorescence profiles were obtained by measuring the AF488 intensity along a straight cross-section drawn manually across the cell and according to the following criteria (S1B Fig): direction = perpendicular to the length of the macrophages, localization = next to the nucleus, width = 30 pixels. Results obtained for 30 to 60 cells were averaged and plotted as a function of the position across the cell.

## Flow cytometry

Cells were seeded at 0,5 x $10^6$ cells/mL in 24-well plates with 1mL per well and stimulated as described above. Macrophages were detached with 10 mM EDTA in PBS for 15 min and were transferred to round bottom 96-well plates, centrifuged at 1000 g for 7 min and then incubated for 35 min on ice in 50 μL of 2 μg/mL FcBlock (anti mouse CD16/CD32) in cytometry buffer (PBS, 0,5% V/V FCS, 2 mM EDTA). Cells were stained with 25 μL of 0.5 μg/mL primary anti mouse-TLR4 (recombinant monoclonal rabbit IgG, clone #1203B, R&D Systems) or isotype (normal polyclonal rabbit IgG control, R&D Systems) in cytometry buffer for 45 min on ice. Cells were then washed by addition of 100 μL of cytometry buffer and centrifuged. The super-natants were discarded before staining with 50 μL of 2 μg/mL secondary AF488 goat anti rabbit (goat IgG H+L, Invivogen) in cytometry buffer for 30 min on ice. 25 μL of fixable viability dye efluor780 (eBioscience) was added for 5 min on ice. Cells were fixed by addition of 50 μL of 2% V/V *para*-formaldehyde for 10 min on ice, washed by addition of 100 μL of cytometry buffer and centrifuged before resuspension in buffer and storage at 4˚C for a maximum period of 24 h before acquisition. Acquisition was done on CytoFLEX (Beckman Coulter) after cali-bration; 10.000 to 30.000 events were recorded for each condition. Analysis was performed using the FlowJo V10 software. The mean fluorescence intensity (MFI) for mTLR4 corre-sponds to live cells.

### Statistical analyses

All the statistical analyses were performed using non-parametric Mann-Whitney test: * p<0.05; ** p<0.01; *** p<0.001. Unless otherwise specified, all experiments were performed with technical triplicates and are representative of at least three independent experiments.

### Ethics statement

All the experiments involving mice conformed with the European Union directive (2010/63 EU) and the French regulation for protection of laboratory animals (February 1, 2013). Project (HA-0036) was approved by the Institut Pasteur ethics committee (CETEA#89).

## Supporting information

**S1 Table. List of primers for RT-qPCR experiments.**
(DOCX)

**S1 Fig. Leptospiral LPS does not trigger TLR4 internalization.**
(TIF)

**S2 Fig. Leptospiral LPS avoids TRIF-dependent responses but activates MyD88.**
(TIF)

**S3 Fig. Shorter leptospiral LPS induces more TLR4-TRIF responses.**
(TIF)

**S4 Fig. Soluble CD14 participates in the enhanced signaling of the shorter leptospiral LPS.**
(TIF)

**S5 Fig. Copurifying lipoproteins contribute to the escape of TLR4 internalization.**
(TIF)

## Acknowledgments

We acknowledge the Center for Translational Science (CRT)—Cytometry and Biomarkers Unit of Technology and Service (CB UTechS) and Photonic BioImaging Unit of Technology and Service (Imagopole) at Institut Pasteur for support in conducting this study.

## Author Contributions

**Conceptualization:** Delphine Bonhomme, Catherine Werts.

**Formal analysis:** Delphine Bonhomme, Ignacio Santecchia, Catherine Werts.

**Funding acquisition:** Ivo G. Boneca, Catherine Werts.

**Investigation:** Delphine Bonhomme, Catherine Werts.

**Methodology:** Delphine Bonhomme, Ignacio Santecchia, Frédérique Vernel-Pauillac, Martine Caroff, Catherine Werts.

**Project administration:** Catherine Werts.

**Resources:** Frédérique Vernel-Pauillac, Pierre Germon, Gerald Murray, Ben Adler.

**Software:** Delphine Bonhomme.

**Supervision:** Catherine Werts.

**Validation:** Delphine Bonhomme, Catherine Werts.

**Visualization:** Delphine Bonhomme.

**Writing – original draft:** Delphine Bonhomme, Catherine Werts.

**Writing – review & editing:** Delphine Bonhomme, Ignacio Santecchia, Frédérique Vernel-Pauillac, Martine Caroff, Pierre Germon, Gerald Murray, Ben Adler, Ivo G. Boneca, Catherine Werts.

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
