## [Decision Letter · Decision Letter 0]

12 Jun 2020

Dear Dr. Werts,

Thank you very much for submitting your manuscript "Leptospiral LPS escapes mouse TLR4 internalization and TRIF‑associated antimicrobial responses through O antigen and associated lipoproteins" for consideration at PLOS Pathogens. As with all papers reviewed by the journal, your manuscript was reviewed by members of the editorial board and by several independent reviewers. The reviewers appreciated the attention to an important topic. Based on the reviews, we are likely to accept this manuscript for publication, providing that you modify the manuscript according to the review recommendations. In addition, for those not familiar with Leptospira  and leptospirosis, I believe that it would be helpful to add a sentence or two to the discussion to proide biological context to your findings, e.g. how this work might affect which host animals are susceptible to severe disease vs. mild disease or asymptomatic carriage.

Sincerely,

Jenifer Coburn, PhD

Associate Editor

PLOS Pathogens

Renée Tsolis

Section Editor

PLOS Pathogens

Kasturi Haldar

Editor-in-Chief

PLOS Pathogens

orcid.org/0000-0001-5065-158X

Michael Malim

Editor-in-Chief

PLOS Pathogens

orcid.org/0000-0002-7699-2064

Reviewer Comments (if any, and for reference):

Reviewer's Responses to Questions

**Part I - Summary**

Reviewer #1: The host response to bacterial lipopolysaccharide (LPS) is mediated via signaling through the pattern-recognition receptor Toll-like receptor 4 (TLR4) via two pathways: a MyD88-dependent route upon recognition of the antigen by TLR4 at the cell surface, and a TRIF-dependent route upon subsequent internalization of LPS-bound TLR4. The authors describe previous work characterizing the unique manner in which the LPS of Leptospira interrogans interacts with the host response, including that although this LPS signals through murine TLR4, it is less endotoxic than the LPS of certain other microbes. In addition, certain lipoproteins that co-purify with leptospiral LPS have been shown to aid in evasion of the early host response. In this report, the authors provide a methodical and thorough investigation of these aspects of the TLR4-mediated host response to LPS of L. interrogans.

Here, the authors show that, unlike the LPS of Escherichia coli, the LPS of a pathological L. interrogans strain did not induce internalization of TLR4. This lack of TLR4 internalization was also observed using the LPS of other L. interrogans serovars, different concentrations of LPS, and different durations of incubation with host cells. High concentrations of leptospiral LPS were shown to less effectively stimulate TRIF-dependent (TLR4-internalized) pathways than LPS of E. coli, leading to less production of nitric oxide (NO) and RANTES, while still stimulating MyD88-dependent pathways. The authors then showed that strains with shorter LPS allowed for more effective internalization of TLR4 and greater production of NO and RANTES than did pathogenic strains with full-length LPS, leading to the hypothesis that possession of full-length LPS allowed for the evasion of TLR4 internalization and the ensuing activation of TRIF-dependent responses. Soluble CD14, a key component of TLR4 signaling, was shown to significantly aid in the production of a TRIF-dependent product in response to the shorter LPS but not to the full-length LPS; the authors hypothesized, after performing binding assays, that the latter effect could be due to a lower affinity between full-length LPS and sCD14. Additionally, leptospiral LPS that was re-purified, with the effect of significantly removing associated lipoproteins, led to increased internalization of TLR4 and increased transcription/production of various TRIF-dependent immune factors. Digestion of LPS with proteinase K led to increased RANTES production, indicating that the protein components of lipoproteins associated with LPS aid in the escape from TRIF signaling. Finally, the findings were supported with a study using whole L. interrograns microbes.

Collectively, these novel findings describe important mechanisms by which L. interrograns can effectively evade critical factors of the host response. The authors’ findings were supported using various experimental methods and were bolstered by robust primary and supplementary data. This was an exceptionally well written and logical manuscript that is suitable for publication with some minor modifications.

Reviewer #2: This is a well-conceived paper by a group of experienced investigators that have a long track record studying the biochemical characteristics of Leptospiral LPS and the host immune response to Lepto LPS in mouse and human cells. They hypothesize that Leptospira LPS, known as a virulence factor, plays a major role in the innate immune evasion of the leptospires, thereby contributing to their stealthy-ness and chronicity in mice. It is a very important study as they provide novel mechanistic data showing that Leptospira LPS escapes the TRIF arm of the TLR4 response to evade RANTES and NO antimicrobial responses. The paper is generally well written and the figures, captions and legends help the reader quickly follow the main objective of the study. One weakness throughout is the lack of clarity between distinction of assessments between mouse and human TLR4. Another is lack of consistency in defining what leads to immune evasion that leads to tolerance and resistance (decreased susceptibility) to infection and evasion from one pro-inflammatory pathway (but not the other) that would lead to increased susceptibility to infection.

Reviewer #3: The manuscript by Bonhomme and colleagues examines cell signaling downstream of the binding of leptospiral LPS to Toll-like receptors with the aim to understand its low endotoxicity. The authors show that leptospiral LPS does not trigger internalization of TLR4, which is potentially linked to its reduced ability to trigger TRIF-dependent responses while MyD88-dependent responses remain intact. Moreover, the group shows that contaminating lipoproteins contribute to the TLR2 response of the LPS but also appear to impair TLR4 internalization and TRIF-dependent responses.

Overall, these are nice findings but there are a few issues to address.

**Part II – Major Issues: Key Experiments Required for Acceptance**

Reviewer #1: 1. At the risk of adding to the abundant data presented here, it was noted that the TRIF-dependent products examined were not presented or described regularly throughout the main figures. It appears that levels of NO were consistently shown, but data on RANTES and, moreso, IL-1beta were not. And, no data on production of IL-1beta protein were shown. A lack of this information does not necessarily negate the main conclusions of this study, but it seems that inclusion of such additional data would further strengthen their conclusions. Would the authors please comment on why these products were not more consistently shown?

2. In Figure 6, the effect of the shorter M895 on NO production was shown, but what about the effect of this microbe on internalization of TLR4?

Reviewer #2: Since there are comparisons between human and mouse TLR4 and TLR2 the authors need to clearly identify throughout the paper what is mouse and what is human. For example, the Abstract and Summary both start with emphasis on human and then end up explaining data seen in mouse. In Discussion it is also important to lead in the first paragraph with the clarification that the data discussed is mouse data (and comparisons and differences w human need to be highlighted throughout).

Reviewer #3: Figure 1: The intensity of the IF stain needs to be increased; it was difficult to see in my copy. Also, please indicated what WGA is staining? While KC induction is fully dependent on MyD88, it’s not clear how much of that signal is TLR4 vs TLR2 dependent. In Figure S2A, the stats are not included for the KC values in the TLR2 and TLR4KO. Did the authors test TLR2/4 double KOs to ensure all of the signal is going through these receptors? Ensure to provide stat tests for mRNA levels in 1B.

The authors also show at low concentrations, leptospiral LPS does not activate either pathway – how does this concentration of LPS relate to an infectious dose? Also, would it be informative to see in vivo with leptospiral LPS injection in mice that it does not induce TRIF-dependent responses? This might hint at importance during in vivo infection.

Figure 3C – Again here the IF was not very easy to see. Also, should include an E coli LPS to compare with the M895 strain.

Figure 3D – Would also be helpful to have comparison with KC; also mRNA levels would add to data.

Figure 4: Western blot for CD14 needs to be improved.

Figure 5D: Again the signal here is difficult to see. G) Stats required.

Figure 6A: For the IF, would be good to have L495 as a comparison.

**Part III – Minor Issues: Editorial and Data Presentation Modifications**

Reviewer #1: 1. Line 141: Figure 1SA doesn’t seem show the data described in this sentence, which describes stimulated cells.

2. Line 147: “cells” needed after RAW264.7.

3. Line 173: Were these cells RAW264.7 cells? The caption for Figure 2B states BMDMs.

4. Line 205: What is meant by “comparable activity” here? The values are different, although the comparison to NS conditions is the same.

5. Line 284: Perhaps “from”, instead of “to”.

6. Line 417: Perhaps “another”, instead of “other”.

7. Line 505: Perhaps “macrophage”, instead of “macrophages”.

8. Lines 835, 837, 893, 907: It is unclear what is meant by “panel” here.

9. Line 839: Missing an open parenthesis.

10. Line 877: “Blot”, rather than “Bot”.

11. Line 963: L495 LPS isn’t mentioned here.

12. Lines 970-974: The description of the dark/light green bars was somewhat confusing. Should the lipase bars in Figures S5B and S5D be light green?

13. Figure S5G: Should “M933” be included in the data label for the LipL32 mutant, as stated in Line 981?

14. Lines 983-984: Wording of “green bars” (plural) to describe each individual group was somewhat confusing.

Reviewer #2: Minor:

Abstract

Line 30-31: needs clarification.

Introduction

In general, this section is very long. Some of this information may be better used in discussion to explain the results observed?

Line 61: I think a more accurate term in this context is enzootic (rather than zoonotic)

Line 72 and 109: consider unconventional instead of peculiar

Line 91: delete interferons (because IFN is a cytokine)

Line 105: consider deleting stricto sensu because it translates into essential

Line 122: consider replacing mystery with unknown

Results

Line 135: after membrane TLR4 add (mTLR4?), it needs to be clarified if mTLR4 stands for membrane or mouse? given that in Fig S3B mTLR4 stands for mouse.

Line 163 (Fig 2): why only 24h? was the analysis not done at 1h also?

Lines 186-188: this setup is a bit confusing, it needs clarification.

Fig 3B: adding the E.coli control here helps visualize the differences

Fig 3C: There not much difference between NS L495 and M895 mTLR4 staining in this figure. This may be easier to visualize against the Ecoli control.

Line 204-206: This comparison needs to be done on the same graph with the same axis. It looks like mTLR4 in LPS Patoc is 5-6X lower than LPS Verdun and hTLR2 is 8X lower?

Line 246: add Fig 5c after treatment (for the silver stain fig)

Fig 5D and 5E: it helps to visualize this against the E. coli control. This is mouse and human. It needs to be clarified

Line 254: replace more produced with increased

Fig 6B: it helps to visualize this against the E. coli control

Line 284: consider replacing participates with contributes

Discussion:

The authors need to better differentiate the use of the term “escape immunity” that lead to tolerance or resistance in mouse from “escape” that leads to susceptibility and severe disease, this becomes quite confusing to the reader. Maybe use escape immunity + stealth of the pathogen, but not escape+virulence of the pathogen.

Discuss the Patoc data.

Line 415: Clarify that this is mouse in the beginning of the sentence.

Methods

HEK293T: clarify that these are human cells.

Where’s your mouse RAW264 macrophages?

Reviewer #3: see above

PLOS authors have the option to publish the peer review history of their article (what does this mean?). If published, this will include your full peer review and any attached files.

Reviewer #1: No

Reviewer #2: No

Reviewer #3: No
---

## [Editor Report · Decision Letter 1]

7 Jul 2020

Dear Dr. Werts,

We are pleased to inform you that your manuscript 'Leptospiral LPS escapes mouse TLR4 internalization and TRIF‑associated antimicrobial responses through O antigen and associated lipoproteins' has been provisionally accepted for publication in PLOS Pathogens.

Best regards,

Jenifer Coburn, PhD

Associate Editor

PLOS Pathogens

Renée Tsolis

Section Editor

PLOS Pathogens

Kasturi Haldar

Editor-in-Chief

PLOS Pathogens

orcid.org/0000-0001-5065-158X

Michael Malim

Editor-in-Chief

PLOS Pathogens

orcid.org/0000-0002-7699-2064
---

## [Editor Report · Acceptance letter]

5 Aug 2020

Dear Dr. Werts,

We are delighted to inform you that your manuscript, "Leptospiral LPS escapes mouse TLR4 internalization and TRIF‑associated antimicrobial responses through O antigen and associated lipoproteins," has been formally accepted for publication in PLOS Pathogens.

Best regards,

Kasturi Haldar

Editor-in-Chief

PLOS Pathogens

orcid.org/0000-0001-5065-158X

Michael Malim

Editor-in-Chief

PLOS Pathogens

orcid.org/0000-0002-7699-2064